# A Feasible Way to Produce Carbon Nanofiber by Electrospinning from Sugarcane Bagasse

**DOI:** 10.3390/polym11121968

**Published:** 2019-11-29

**Authors:** Wei Chen, Xin-Tong Meng, Hui-Hui Wang, Xue-Qin Zhang, Yi Wei, Zeng-Yong Li, Di Li, Ai-Ping Zhang, Chuan-Fu Liu

**Affiliations:** 1State Key Laboratory of Pulp and Paper Engineering, South China University of Technology, Guangzhou 510640, China; geogeo_chen@163.com (W.C.); mxt08205200@163.com (X.-T.M.); wang.huihui@mail.scut.edu.cn (H.-H.W.); zhangxueqin0228@163.com (X.-Q.Z.); wygift@163.com (Y.W.); 13609763220@163.com (Z.-Y.L.); lhnlg0503@126.com (D.L.); 2College of Light Industry and Food Science, Zhongkai University of Agriculture and Engineering, Guangzhou 510225, China; 3College of Forestry and Landscape Architecture, South China Agricultural University, Guangzhou 510642, China; aiping@scau.edu

**Keywords:** electrospinning, homogeneous esterification, acid anhydrides, carbon nanofiber, lignocellulosic biomass

## Abstract

Recently, the nanofiber materials derived from natural polymers instead of petroleum-based polymers by electrospinning have aroused a great deal of interests. The lignocellulosic biomass could not be electrospun into nanofiber directly due to its poor solubility. Here, sugarcane bagasse (SCB) was subjected to the homogeneous esterification with different anhydrides, and the corresponding esterified products (SCB-A) were obtained. It was found that the bead-free and uniform nanofibers were obtained via electrospinning even when the mass fraction of acetylated SCB was 70%. According to the thermogravimetric analyses, the addition of SCB-A could improve the thermal stability of the electrospun composite nanofibers. More importantly, in contrast to the pure polyacrylonitrile (PAN) based carbon nanofiber, the SCB-A based carbon nanofibers had higher electrical conductivity and the surface N element content. In addition, the superfine carbon nanofiber mats with minimum average diameter of 117.0 ± 13.7 nm derived from SCB-A were obtained, which results in a larger Brunauer–Emmett–Teller (BET) surface area than pure PAN based carbon nanofiber. These results demonstrated that the combination of the homogeneous esterification and electrospinning could be a feasible and potential way to produce the bio-based carbon nanofibers directly from lignocellulosic without component separation.

## 1. Introduction

Electrospinning is a simple and multifunctional technique for producing continuous nanofibers [1,2]. It has received lots of attention due to the extensive sources of raw materials, such as synthetic and natural polymers, polymer alloys, polymers loaded with chromophores, nanoparticles, active agents, metals and ceramics, and so on [3,4,5]. Commonly, the fossil-based polymer is a preferred precursor of electrospinning due to their good solubility and spinnability. However, the excessive consumption of fossil resources caused the rise in prices, which has greatly limited its application [6]. Thus, in view of the economy and environmental protection, replacing partial or the entire fossil-based polymer with renewable polymer, such as cellulose, lignin, and chitosan, is an irresistible trend. 

Lignocellulose biomass, an abundant renewable resource including plants, wood, and agricultural residues, can be used for producing biomaterials, bioenergy, and biochemical products [7,8,9]. It is composed of cellulose, hemicellulose, and lignin with different ratios. Cellulose is the linear polymer component and has good chemical resistance, thermal stability, and biodegradability [10,11]. Many literatures reported the spinning of nanofiber with cellulose or its derivatives, including microcrystalline cellulose, bacterial cellulose, cellulose acetate, and so on [12,13,14]. As a highly abundant biomaterial with high carbon content and byproduct of the pulping and cellulosic ethanol industries [15,16], lignin has also explored to produce electrospinning nanofibers. Thus, it is reasonable to regard the lignocellulose as a potential and suitable replacement of the electrospinning precursor. 

However, it is impossible to electrospun lignocellulose directly due to its complicated chemical structures and complex chemical linkages between different components [17]. The insolubility of lignocellulose in the conventional solvents is the biggest obstacle for its application on electrospinning [18]. The solubility of lignocellulose was improved for the direct utilization of all components after derivatization, especially the esterification reaction [18,19,20]. In general, the esterification reaction of lignocellulose is mainly divided into heterogeneous and homogeneous processes [18,20]. Compared with the homogeneous esterification, the heterogeneous esterification of lignocellulose is incomplete and inefficient because the esterification mainly occurred on the highly lignified cell corner and compound middle lamella [21,22]. In addition, the heterogeneous esterification has many disadvantages, including side reactions, longer time, corrosion, and obvious degradation [23,24]. Therefore, it is believed that the homogeneous process is comparatively a better choice for the esterification reaction of lignocellulose.

In this study, a new strategy was proposed to produce the carbon nanofibers by carbonization of the nanofiber mats electrospun directly from sugarcane bagasse (SCB) without fractionation after the homogeneous esterification with different anhydrides. The esterification of SCB was investigated with Fourier transform infrared (FT-IR), and ^1^H and ^13^C nuclear magnetic resonance (NMR). The spinnability of SCB esters (SCB-A) was evaluated with the characterization of viscosity and electric conductivity of electrospinning solutions and the surface morphology analysis of different as-spun nanofiber mats with scanning electron microscope (SEM). The properties of the carbonized nanofibers were explored with X-ray diffraction (XRD), Raman spectrometer, and X-ray photoelectron spectroscopy (XPS) analyses.

## 2. Materials and Methods

### 2.1. Materials

The SCB was collected from a local factory in Jiangmen of China, air-dried, and cut into small pieces. After that, the fraction was ground to 40–60 mesh size (450–900 μm) powder (220 rpm for 4 h) with Planetary Micro Mill (Fritsch, Idar-Oberstein, Germany). PAN (*M*w = 150,000 g/mol) powder was purchased from Macklin Reagent Co. (Shanghai, China). Acetic anhydride (AA, 98.5%), propionic anhydride (PA, 98.5%), butyric anhydride (BA, 98.0%), and *N,N*-dimethylformamide (DMF, 99.8%) were purchased from Aladdin Chemistry Co. Ltd. (Shanghai, China). All chemicals used in this study were of analytical grade.

### 2.2. Homogeneous Esterification of SCB

In order to eliminate the influence of extracts, the SCB powder was extracted with toluene-ethanol (2:1, v/v) for 4 h, and then air-dried at 50 °C for 24 h. Typically, 0.5 g of the extractive-free SCB powder was dispersed in dimethyl sulfoxide (DMSO)/*N*-methylimidazole (NMI) (2:1, v/v) dual solvent system (10.0 g) at room temperature with agitation for 10 min and then heated to 90 °C with stirring for 4 h to obtain a clear solution. After that, 40 mmol/g of anhydride (the ratio of anhydride to SCB) was slowly added to the clear solution, and the mixture was continuously agitated at 90 °C for 90 min for esterification reaction. All the reaction processes were conducted under nitrogen atmosphere. After the required time, the resulted mixture was cooled to room temperature and added into ethanol (99.0 wt%, 200 mL) with agitation. The esterified SCB products were filtered and washed with ethanol (four times, total 800 mL) to remove DMSO, NMI, residual anhydride, and other impurities, and then freeze-dried. The obtained solid residues esterified with acetic anhydride, propionic anhydride, and butyric anhydride were denoted as SCB-AA, SCB-PA, and SCB-BA, respectively, and collectively referred as SCB-A. Meantime, the successful esterification of SCB with three different acid anhydrides was confirmed by the FT-IR and NMR analyses (Appendix A).

### 2.3. Preparation of SCB-A/PAN Blends Electrospun Nanofiber Mats (SCB-A/PAN-SNFs)

In this study, in order to increase the spinnability of SCB-A, a certain amount of PAN was added to the spinning solution. The SCB-A/PAN blends, the mass fraction of SCB-A range from 50% to 90%, were dissolved in DMF and stirred for 12 h under 50 °C to obtain the precursor solution with a solid concentration of 12 wt %. The solution was filled in a syringe equipped with a 21 G metallic needle, which was connected to a high-voltage power supply, and a rectangular steel plate covered with aluminum foil was used as a collector. The high voltage, feeding rate, temperature, relative humidity, and distance between the anode and cathode were fixed at 10 kV, 0.5 mL/h, 30 °C, 30%, and 10 cm, respectively. Then, the SCB-A/PAN-SNFs were dried at 50 °C in a vacuum oven overnight. 

### 2.4. Heat Treatment of the Electrospun Nanofiber Mats

The PAN based nanofiber mats (PAN-SNF) and SCB-A/PAN-SNFs were thermally treated to convert them into carbonaceous materials. Thermo stabilization was performed by heating the samples at 0.5 °C/min to 240 °C in an air atmosphere and holding for 4 h. The stabilized nanofiber mats were subsequently carbonized by heating samples at 5 °C/min to 800 °C and holding for 2 h. Schematic description of the preparation of superfine carbon nanofiber mats derived from sugarcane bagasse is summarized in Figure 1.

In this study, the obtained SCB-A/PAN-SNFs were named in the form of “AA-SNF-50%”. The “AA” represented the “SCB-AA”, “SNF” was the abbreviation of electrospun nanofibers, and the “50%” represented that mass fraction of the SCB-A was 50%. Similarly, the stabilized electrospun nanofiber mats (SCB-A/PAN-TNFs) were named as “AA-TNF-50%”, and the carbonized electrospun nanofiber mats (SCB-A/PAN-CNFs) were named as “AA-CNF-50%”. The nanofiber mats obtained from pure PAN were named as PAN-SNF, PAN-TNF, and PAN-CNF. 

### 2.5. Characterization

FT-IR was recorded on an FT-IR spectrophotometer (Bruker, Karlsruhe, Germany) using the KBr disk method in the range of 4000−400 cm^−1^. The ^1^H NMR and ^13^C NMR spectra were recorded from 40 mg samples in 0.5 mL of DMSO-*d*_6_ on a Bruker Avance III 400 M spectrometer (Bruker, Karlsruhe, Germany).

The viscosity and electric conductivity of different SCB-A/PAN blend solutions were measured with cone-and-plate viscometer (RM First Plus, Lamy Rheology Instruments, Arrondissement de Lyon, France) in a constant shear rate of 100 s^−1^ and electrical conductivity meter (Five Easy Plus FE38, Mettler Toledo, Zurich, Switzerland) respectively. The temperature of the solutions was kept at 25 °C and every reading was recorded until reaching equilibrium. 

The surface morphology of different nanofiber mats was analyzed using SEM (ZEISS Merlin, Oberkochen, Germany) at an accelerating voltage of 10 kV. The thermogravimetric and derivative thermogravimetry analyses (TGA/DTG) were performed on a TGA Q500 thermogravimetric analyzer (TA, New Castle, DE, USA) at a heating rate of 20 °C/min to 700 °C in an inert atmosphere. Elemental compositions (percentage of C, H, N and S) of the different nanofiber mats were determined using an elemental analyzer of Vario EL Cube apparatus (Elementar, Langenselbold, Germany).

The electric resistivity of CNFs derived from different as-spun nanofiber mats was measured by a handheld four-point probe (M-3, Suzhou Jingge Electronic Co., Ltd., Suzhou, China). The surface area of carbon nanofibers was determined from N_2_ (77.4 K) adsorption–desorption isotherms using an ASAP-2046 surface area analyzer. The surface area was calculated using a Brunauer–Emmett–Teller (BET) method in the linear range of P/P_0_ = 0.01–0.1. Samples were degassed at 150 °C for 3 h under vacuum before measurement. 

The crystallinity of samples was characterized by XRD measurement (D8 ADVANCE, Bruker, Cu Kα radiation, λ = 1.54 Å). The acceleration voltage and emission current were 40 kV and 40 mA, respectively. The interplanar spacing (*d*_002_), the lateral size (*L*_*a*_, also known as the in-plane crystal size) and the crystallite size (*L*_*c*_) of the different CNFs were calculated using the Bragg’s law (Equation (1)) and Debye-Scherrer equations (Equations (2) and (3)):(1)$$d_{002}=\frac{\lambda }{2sin\theta },\;$$
(2)$$L_{a}=\frac{1.84\lambda }{\beta \left(100\right)cos\theta \left(100\right)},$$
(3)$$L_{c}=\frac{0.89\lambda }{\beta \left(002\right)cos\theta \left(002\right)},\;$$
where 𝜃 and 𝛽 are the diffraction angle and the full width at half maximum (FWHM) of diffraction peaks in radians, respectively. 

The ratio of graphitic and amorphous carbons was determined using a LabRAM Aramis Raman spectrometer equipped with a 633 nm He–Ne laser. The XPS analysis was performed on a Kratos Axis Ulra DLD spectrometer under a pressure of 5 × 10^−9^ torr. The XPS spectra were acquired using monochromatic Al Kα X-ray source (1486.6 eV) at 5 mA × 15 KV over an area of 700 × 300 µm^2^ at an incident angle of 45°. All binding energies were referenced to adventitious C 1s at 284.6 eV. Chemical states of elements were assigned based on the National Institute of Standards and Technology (NIST) XPS Database. 

## 3. Results and Discussion

### 3.1. Spinnability and Rheology of SCB-A/PAN Blends Solutions

In order to obtain high-quality nanofibers, it is crucial that bead-free, uniform, and fine precursor nanofibers were required. There are many parameters that could affect the electrospinning process. In addition to instrumental parameters (flow rate, applied voltage, and distance between tip and collector) and ambient parameters (temperature and humidity), the solution parameters (viscosity and conductivity) could directly influence the spinnability of the precursor solution [2,25,26]. Thus, the solution properties (viscosity and conductivity) of different precursor solutions were investigated, as shown in Figure 2. 

The conductivity of SCB-A/PAN blends solutions decreased from 103.5 to 58.4 μS/cm, 133.3 to 65.7 μS/cm, and 137.4 to 71.1 μS/cm as the mass fraction of SCB-AA, SCB-PA, and SCB-BA increased from 50% to 90%, respectively. It indicated that the addition of SCB-A had a negative influence on the overall conductivity of the precursor solutions. Meantime, compared to the SCB-AA, the conductivity of precursor solutions based on SCB-PA and SCB-BA increased gradually at the same mass fraction. 

The SEM analysis of the SCB-A/PAN-SNFs with different mass fractions of SCB-A was conducted, and the corresponding SEM images and average diameters of electrospun nanofibers are shown in Figure 3. As we can see that the uniform nanofibers were formed when the mass fraction of SCB-A was 50%, and their average diameters were 149.3 ± 32.1 nm (AA-SNF-50%), 158.2 ± 37.2 nm (PA-SNF-50%), and 160.1 ± 26.0 nm (BA-SNF-50%), respectively. At the same concentration of 12 wt %, the PAN-SNF had much larger average diameter of 502.9 ± 50.6 nm (Appendix A). When the mass fraction of SCB-A increased up to 70%, the occasional and few beads emerged on the PA-SNF-70% and BA-SNF-70%, while no beads observed on the AA-SNF-70%. The further addition of 90% SCB-A into precursor solution resulted in an increasing density of beads and forming the beaded structure nanofiber. 

For the most part, the diameters and uniformity of SCB-A/PAN-SNFs were declined with increasing the mass fraction of SCB-A, owing to the decline of the conductivity and viscosity of the precursor solutions (Figure 2). As previously reported, the continuous and uniform nanofiber cannot be obtained when the viscosity was extremely low [27]. Actually, the improvement of solutions conductivity would led to the increase of static charge density of the jet, and then a strong elongation forces are imposed to the polymers formed a bead-free and uniform nanofiber [27,28,29]. In general, the standard deviation of nanofiber average diameters, as well as the nanofiber morphology, indicated that the SCB-AA had a better electrospinning performance than SCB-PA and SCB-BA. The bead-free and uniform nanofibers were obtained even when the mass fraction of SCB-AA was 70%.

### 3.2. Pyrolysis Behavior of SCB-A and the Corresponding Nanofiber Mats

The TGA/DTG analysis was used to evaluate the thermal properties of SCB-A and the corresponding nanofiber mats. The TGA/DTG curves and the pyrolysis characteristics of different samples were obtained (Figure 4 and Appendix A). 

As we can see that the main DTG thermograph peaks were observed in the zone of temperature 200–500 °C, which was related to the decomposition of hemicellulose and cellulose. As shown in Figure 4a and Appendix A, the SCB was fairly stable up to 275.8 °C without significant weight loss, while its derivatives (SCB-PA and SCB-BA, except for SCB-AA) showed higher initial decomposition temperature (T_in_) and maximum decomposition temperature (T_max_). The poor thermal stability of SCB-AA may be due to the decrease of abundant hydroxyl groups after esterification reaction, which led to the breakage of the hydrogen bonding network of SCB [30]. Furthermore, the degradation of SCB during the homogenous esterification could also be responsible for the decrease of the thermal stability of SCB-AA. The T_in_ of SCB-A increased from 272.2 to 292.2 °C along with the increasing chain length of acyl group, and the same trend can be observed from results of T_50_ and T_max_. These results indicated that the increase in the chains of acyl groups could improve the thermostability of SCB-A, which were consistent with previous researches [31,32]. 

In contrast, pure PAN-SNF showed higher T_in_ and T_max_ than SCB-A, which was probably related with the high molecular of PAN (Figure 4b). The SCB-A showed a wider degradation region compared with PAN-SNF due to its complex compositions. According to the previous report [33], the thermal behaviors for the co-pyrolysis of mixture components were more complex than that of the individual component. After the SCB-A blended electrospun with PAN, the thermal stability of SCB-A/PAN-SNFs was improved even better than PAN-SNF. For example, the main weight loss of AA-SNF-50% and BA-SNF-50% were recorded at 328.1 and 331.3 °C, both higher than that of pure PAN-SNF by 16.8 and 20.0 °C, respectively. Similar phenomena have also been reported in previous research papers [33,34], which were known as the positive synergistic interaction between biomass and plastics during the pyrolysis. As shown in Figure 4a, the residues of SCB-AA, SCB-PA, and SCB-BA were 24.8%, 17.6%, and 17.1%, respectively, which were much less than that of pure PAN-SNF (Figure 4b). However, after the blend electrospun with PAN, the residue of AA-SNF-50% was equal to that of PAN-SNF (36.1%), while the PA-SNF-50% and BA-SNF-50% had a relative lower residue at 700 °C. In general, the findings of this part demonstrates that the electrospinning of SCB-A/PAN blends resulted in the improvement of thermal stability of as-spun nanofiber compared to individual component (PAN or SCB-A) due to the existence of synergistic interaction.

### 3.3. Thermo Stabilization and Carbonization of SCB-A/PAN-SNFs

In this study, the PAN-SNF and SCB-A/PAN-SNFs with the SCB-A mass fraction of 50% were subjected to thermo stabilization and carbonization, and then the corresponding nanofiber mats were obtained (Appendix A and Figure 5). It can be seen that the average diameter of pure PAN based nanofiber at the different stages were 396.3 ± 53.7 nm (PAN-TNF) and 254.9 ± 37.7 nm (PAN-CNF), respectively. After the thermo stabilization process, the nanofiber average diameters were decreased obviously during the pyrolysis stage. Meanwhile, it was found that the average diameter of SCB-BA based nanofiber was decreased from 152.6 ± 27.8 nm (BA-TNF-50%) to 117.0 ± 13.7 nm (BA-CNF-50%; Figure 5). At the same electrospinning conditions, the SCB-A/PAN-CNFs had smaller average diameters and about two times higher BET surface area than that of PAN-CNF (Table 1). According to the previous researches [35,36], the decline of the fiber average diameters was responsible for the improvement of the BET surface area. 

XRD analysis (Figure 6a and Appendix A) and Raman spectroscopy (Figure 6b) were used to characterize the graphitization structure of different electrospun CNFs. Two peaks located at 2θ = 24° (major) and 43° (minor) can be indexed to typical crystallographic plane of (002) and (100) graphitic carbon, respectively [37,38]. As shown in Appendix A, the graphitic interplanar spacing (*d*_(002)_) of different CNFs ranged from 3.61 to 3.64 Å, the crystallite size (*L_c_*) ranged from 0.94 to 1.02 nm, and the lateral size (*L_a_*) ranged from 2.65 to 3.51 nm. Those small changes of XRD physical characterization indicated that the SCB-A/PAN-CNFs had a similar degree of graphitization with PAN-SNF. As shown in Figure 6b, the Raman spectroscopies of different samples present the obvious structure of carbonaceous materials. The two strong peaks at 1350 and 1598 cm^−1^ were assigned to the defect or disordered sp^2^ carbon (D) band and the E_2g_ graphite (G) band, respectively [37]. The intensity ratio of D peak and G peak (I_D_/I_G_; known as the “R-value’’) indicated the degree of graphitization of the carbonaceous materials [37,39]. The R-value ranged from 1.06 (PAN-CNF) to 1.01 (BA-CNF-50%), indicating that their graphitization degrees were identical. Therefore, the XRD and Raman spectroscopy results demonstrated that the addition of SCB-A would not result in a decrease of graphitization degree of corresponding CNF.

The surface valence and chemical compositions of these CNFs were further investigated by XPS (Figure 6c), and the contents of different elements (C, N, and O) on the nanofiber surface were present in Table 1. The curve-fitted high-resolution XPS N 1s spectra were obtained with the software of XPS Peak Fit (Figure 6d). There were three distinguished nitrogen species in N 1s spectra of SCB-A/PAN-CNFs, including pyridinic-N (398.0 eV), pyrrolic-N (399.5 eV), and graphitic-N (401.0 eV), while no pyrrolic-N was showed in N 1s spectra of PAN-CNF. It was believed that pyridinic and pyrrolic N species can contribute to the accumulation of electric charge during the charge–discharge process due to the appropriate electron configuration and binding energy [40,41]. Meantime, the doping of pyrrolic-N and pyridinic-N can not only improve the conductivity, but also change the electronic state of the surface carbon of the material [42]. The electrical conductivity of different CNFs was determinated with four-point probe method and the results were shown in Table 1. It was found that the SCB-A/PAN-CNFs had better electrical conductivity than PAN-CNF, which could attribute to its higher content of pyrrolic-N and pyridinic-N.

In contrast to XPS analysis, the elemental analysis was an overall chemical composition analysis method. As shown in Table 1, the C content of different CNFs was consistent without large variations, while the surface C content (about 86%) was higher than that of overall C content (about 75%). The increased relative surface C content was attributed to the decline of N and O elements contents, which means that the carbonization process was carried out from the surface to the inside of the fiber. Meanwhile, the surface O content of SCB-A/PAN-CNFs was much lower than the overall O content. It meant that the element O on the surface of SCB-A/PAN-CNFs could be removed more easily than element N in an inert atmosphere. Moreover, it is worth noticing that, unlike the C content, the N content of AA-CNF-50% (8.22%), PA-CNF-50% (9.96%), and BA-CNF-50% (8.80%) were twice more than that of PAN-CNF (3.80%). Thus, these results indicated that the blend electrospun of SCB-A with PAN might be beneficial to reserve the surface N atoms of carbon nanofibers during the pyrolysis carbonization stage. It will favor the production of carbon material defects due to the fact that doping with heteroatoms is an enabling strategy to improve electrochemical reactivity and electrical conductivity of carbonaceous materials [43,44]. Therefore, according to the results above, it was believed that SCB-A/PAN-CNFs could be considered as potential electrochemical material as its superior electrical conductivity and high surface activity (high surface N content and large surface area).

## 4. Conclusions

In summary, the aliphatic side chains were grafted onto SCB by esterification successfully with different acid anhydrides. During the electrospinning process, the SCB-AA showed a better spinnability than SCB-PA and SCB-BA. Moreover, the SCB-A/PAN-CNFs had small mean diameters, which resulted in a higher BET surface area. The TGA/DTG analysis indicated the thermal stability of as-spun nanofiber had been improved with the addition of SCB-A even better than that of pure PAN-SNF due to the synergistic interaction between PAN and SCB-A. Overall, this study provides a feasible way to employ the lignocellulose without component separation for the production of nanofiber materials. Moreover, the SCB-A/PAN-CNFs provides some interesting properties compared to PAN-CNF, such as small diameters, high surface N content, and large surface area. Therefore, it is suggested that SCB-A/PAN-CNFs can be considered as potential conductive templates in electrochemical energy storage and conversion devices.

## Figures and Tables

**Figure 1 polymers-11-01968-f001:**
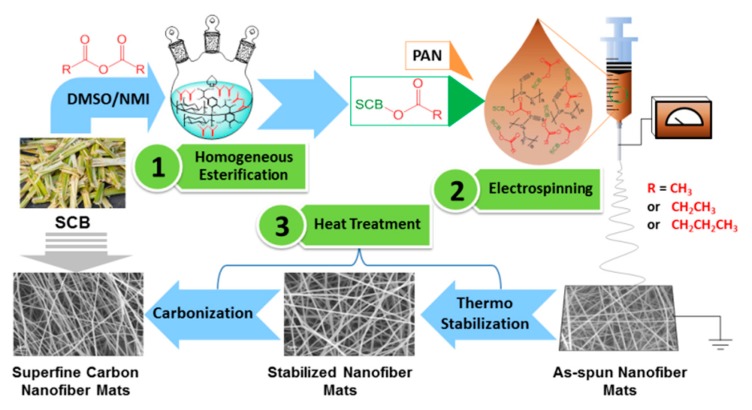
Schematic description of the preparation of superfine carbon nanofiber mats derived from sugarcane bagasse.

**Figure 2 polymers-11-01968-f002:**
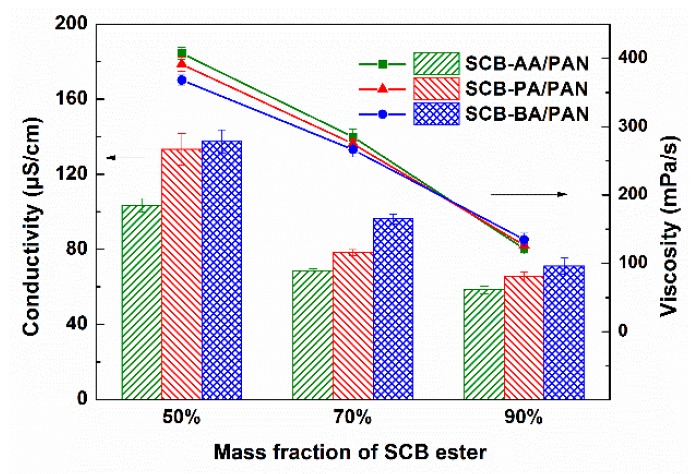
The conductivity and viscosity of sugarcane bagasse esters (SCB-A)/PAN blend solutions with different mass fraction of SCB esters.

**Figure 3 polymers-11-01968-f003:**
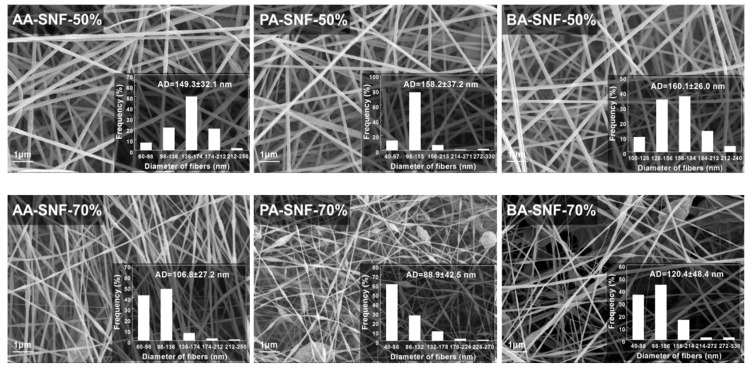
SEM images of SCB-A/PAN blend electrospun nanofiber mats with different mass fractions of SCB esters (scale bar = 1 μm).

**Figure 4 polymers-11-01968-f004:**
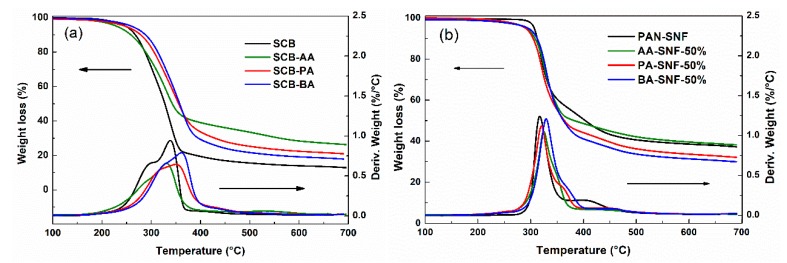
TGA/ derivative thermogravimetry analyses (DTG) analysis of (**a**) SCB and different SCB esters and (**b**) PAN and SCB-A/PAN blend electrospun nanofiber mats.

**Figure 5 polymers-11-01968-f005:**
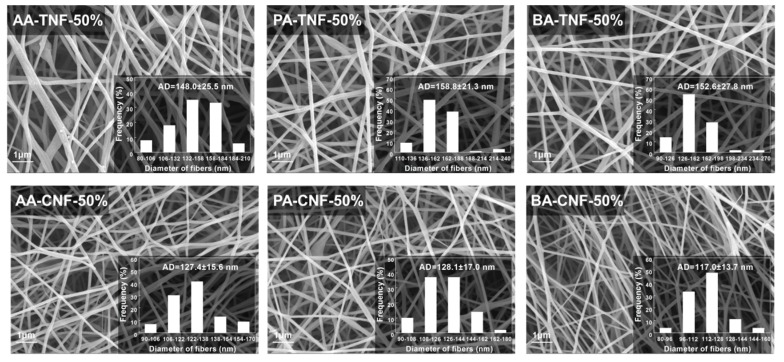
SEM images of different SCB-A/PAN-stabilized electrospun nanofiber mats (TNFs) and SCB-A/PAN-carbonized electrospun nanofiber mats (CNFs; scale bar = 1 μm).

**Figure 6 polymers-11-01968-f006:**
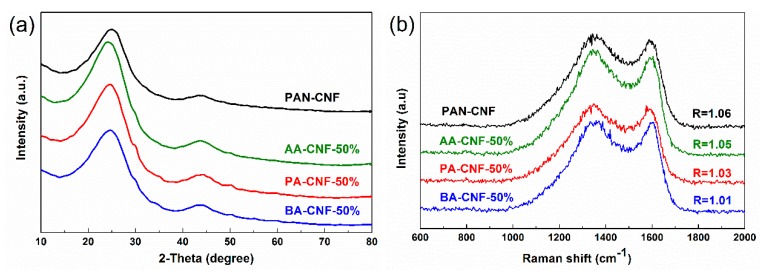
(**a**) XRD spectra; (**b**) Raman spectra; (**c**) general XPS spectra; and (**d**) XPS deconvoluted N 1s spectra of different carbon nanofiber mats.

**Table 1 polymers-11-01968-t001:** The elemental analysis, X-ray photoelectron spectroscopy (XPS) analysis, electrical conductivity, and Brunauer–Emmett–Teller (BET) surface area of different CNFs.

Sample	Elemental Analysis			XPS Analysis			EC (S/cm) ^2^	AD (nm) ^3^	S_BET_ (m^2^/g) ^4^
	C (%)	N (%)	O (%) ^1^	C (%)	N (%)	O (%)			
PAN-CNF	74.2	15.4	8.70	87.8	3.80	8.42	8.97 × 10^−5^	254.9 ± 37.7	12.9
AA-CNF-50%	75.7	11.1	11.7	86.9	8.22	4.86	2.18 × 10^−4^	127.4 ± 15.6	36.8
PA-CNF-50%	74.5	11.8	13.2	86.0	9.96	4.35	2.49 × 10^−4^	128.1 ± 17.0	35.8
BA-CNF-50%	75.4	11.0	12.5	86.7	8.80	4.47	3.71 × 10^−4^	117.0 ± 13.7	34.6

^1^ O % of elemental analysis was determined by subtraction of C %, H %, and N % from the total composition; ^2^ EC: Electrical conductivity; ^3^ AD: Average diameter; ^4^ S_BET_: BET surface area.

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
