# Peer review of "A Feasible Way to Produce Carbon Nanofiber by Electrospinning from Sugarcane Bagasse"

_polymers, 2019, doi:10.3390/polym11121968_

Round 1
Reviewer 1 Report
Comments on the paper “A Feasible Way to Produce Carbon Nanofiber by Electrospinning from Sugarcane Bagasse” by Wei Chen et al.
The paper reports the preparation of carbon nanofibers by the chemical modification of an agricultural waste which is further blended with PAN. This mixtures are then electrospun and the resulting fibers are carbonized. The subject of the paper is not new as there are many papers dealing with this issue i.e. with the use of PAN blended with several chemical in order to produce carbon microfibers. Moreover, the paper does not show new insights.
The BET surface areas of the carbon fibers are reported in table 2 (page 9, line 326) but the procedure is not included in the experimental section.
The XPS experimental conditions are not reported in the experimental section. Moreover no information on the peak fitting procedure (page 9) is supplied.
3.1 Section should be included in Supporting Information. Moreover the interest of table 1 is really scarce so that it can be eliminated.
Line 231. “…..the uniform nanofibers were formed when the mass fraction of SCB-A was 50%,…”. Nevertheless, BA-SNF-50% shows a size distribution which is similar or even less uniform than these of -SNF-70% series.
Please note, figure 6 shows the samples have very similar weight losses. Thus the discussion in lines 263-293 is of scarce interest.
Similarly the XRD and Raman spectra of the samples are very similar (figures 8a and b). Therefore nothing can be deduced from them. Please note the ID/IG ratios are almost coincident (1.01-1.06) as the tiny differences are between the experimental error.
Table 2. The differences between the elemental analysis and the XPS data should be explained. Moreover, the electrical conductivity of BA-CNF-50% is clearly larger than the other samples, but no explanation is supplied.
English needs editing.
To sum up: in my opinion this paper does not deserve publication.
Author Response
Dear Reviewer,
We would like to thanks for your thoughtful review of our manuscript. Those comments are all valuable and very helpful for revising and improving our paper, as well as the important guiding significance to our researches.
The main correction in the paper and the responds to the comments are list in the Word file.
Please see the attachment.
Once again, thank you very much for your comments and suggestions.
Best wishes.
Correspondence author: Chuan-Fu Liu

Reviewer 2 Report
In this study, CNFs were prepared by blends of PAN and pre-treated natural products. Addition of SCBs provides some interesting properties such small diameters of electrospun nanofibers, and high N content and large surface area of CNFs. For many applications, these properties are desirable and thus these CNFs seem to have many potential applications.
1. Have the author synthesized carbon using esterified SCBs only (it doesn't need to be a nanofiber form)? The large surface area of CNFs derived from blends can be due to the thermal decomposition of SCBs during thermal treatments. So it would be helpful to measure the carbon yields or Raman spectroscopy of carbon derived from SCBs only to make it clear.
Author Response

(The authors gave the same response as above.)

Reviewer 3 Report
The manuscript details the preparation and characterization of carbon fiber electrospun fibers derived from sugarcane bagasse . The topic is timely and of interest to a broad audience including polymer chemists.
Moreover, the subject of the manuscript fits the scope of the journal. However, there are several issues that must be addressed prior to acceptance to be published in Polymers:
Authors should discuss about possible fields of applications of the developed carbon nanofibers Please describe the parameters of the ball milling process The role of PAN in not described in the manuscipt Authors should consider the addition of salts to adjust the conductivity of the polymeric solutionAuthor Response
Dear Reviewer,
We would like to thanks for your thoughtful review of our manuscript. Those comments are all valuable and very helpful for revising and improving our paper, as well as the important guiding significance to our researches.
The main correction in the paper and the responds to the comments are list in the Word file. Please see the attachment.
Once again, thank you very much for your comments and suggestions.
Best wishes.
Correspondence author: Chuan-Fu Liu

Round 2
Reviewer 1 Report
Comments about the paper “A Feasible Way to Produce Carbon Nanofiber by Electrospinning from Sugarcane Bagasse” by Wei Chen et al.
The experimental conditions to obtain de surface areas (degasification, time and temperature) are still messing.
Similarly the experimental conditions to obtain the survey and high resolution XPS spectra as well as the procedure to fit the spectra are not supplied.
I still think that the paper should be shortened by including 3.1 section in Supporting Information. I also think the interest of table 1 is really scarce so that it can be eliminated.
My previous comment about the XRD and Raman was that the spectra of the samples are very similar (figures 8a and b). I still think that nothing can be deduced from them. The authors have sent some references in which the values of the ID/IG ratios are similar. The fact that similar values have been reported does not mean they allow concluding something about changes in the textural characteristics. I still think the ID/IG ratios are almost coincident (1.01-1.06) and they are between the experimental errors.
Please note that still English needs editing.
Author Response
Dear Reviewer,
We would like to thanks for your thoughtful review of our manuscript. Those comments are all valuable and very helpful for revising and improving our paper, as well as the important guiding significance to our researches.
The main correction in the paper and the responds to the comments are list in the Word file. Please see the attachment.
Once again, thank you very much for your comments and suggestions.
Best wishes.
Corresponding author: Prof. Chuan-Fu Liu

Reviewer 3 Report
The authors have improved the quality of the paper and the manuscript has presented in acceptable format.
I would encourage the authors to conduct a final grammatical review of the manuscript
Author Response

(The authors gave the same response as above.)

Round 3
Reviewer 1 Report
The paper can be accepted in its present form.